DATA RELEASE

# Transcriptomic profile of embryoid bodies under hypoxia at single cell level

Bárbara Acosta-Iborra[1,†], Yosra Berrouayel[1,*,†], Laura Puente-Santamaría[1], Luis del Peso[1,2,3,4] and Benilde Jiménez[1,2,3,4]

1 Departamento de Bioquímica, Universidad Autónoma de Madrid (UAM), Instituto de Investigaciones Biomédicas "Sols-Morreale" (CSIC-UAM), Madrid, Spain
2 IdiPaz, Instituto de Investigación Sanitaria del Hospital Universitario La Paz, Madrid, Spain
3 Centro de Investigación Biomédica en Red de Enfermedades Respiratorias (CIBERES), Instituto de Salud Carlos III, Spain
4 Unidad Asociada de Biomedicina, CSIC, Universidad de Castilla - La Mancha, Spain

## ABSTRACT

Oxygen availability is a key regulator of cellular physiology and hypoxia plays a central role driving vasculogenesis and angiogenesis during development. Although bulk transcriptomics has revealed important oxygen-regulated gene networks, such approaches cannot resolve the cellular heterogeneity and lineage dynamics characteristic of early differentiation. To address this, we generated a single-cell transcriptomic dataset from murine embryoid bodies, a widely used *in vitro* model of early embryonic development, cultured 8 or 10 days under hypoxic (1% $O_2$) or normoxic (21% $O_2$) conditions for the final 16 or 48 hours of differentiation. This resource enables detailed exploration of how oxygen availability influences lineage specification, vascular and hematopoietic development, and cellular heterogeneity during early differentiation. Beyond developmental biology, the dataset provides a valuable reference for comparative studies of hypoxia responses, benchmarking of single-cell analysis methods, and integrative investigations into oxygen signaling across diverse biological systems.

**Subjects** Genetics and Genomics, Transcriptomics, Bioinformatics

**Submitted:** 11 November 2025

\* Corresponding author. E-mail: yberrouayel@iib.uam.es

† Contributed equally.

Preprint submitted at https://doi.org/10.1101/2025.10.29.685315

## DATA DESCRIPTION

### Context

Oxygen is a central regulator of cellular metabolism, with hypoxia playing a crucial role in tissue formation and vascular growth during embryogenesis [1]. Here, we present single-cell transcriptomic profiles of mouse embryoid bodies differentiated under controlled oxygen conditions. The dataset comprises thousands of cells collected at two differentiation stages, following either short or prolonged hypoxic exposure, enabling direct comparisons across four experimental conditions. These profiles capture a broad spectrum of developmental states, from progenitors to more differentiated lineages, providing a detailed view of how oxygen availability shapes lineage specification, vascular and hematopoietic development, and cellular heterogeneity. The original aim included examining the generation and expansion of endothelial cells under hypoxia. However, mature endothelial cells were relatively rare in these samples, precluding a robust assessment of our hypothesis. Nonetheless, the dataset captures a broad spectrum of other

**Figure 1.** Experimental design. Schematic representation of the experimental design. Mouse embryonic stem cells were differentiated into embryoid bodies for 8 or 10 days under normoxic or hypoxic conditions, with hypoxia applied for 16 h (day 8) or 48 h (day 10).

cell populations, including mesodermal progenitors, endodermal precursors, and more differentiated lineages. Beyond its immediate biological insights, the dataset offers a valuable resource for comparative analyses of differentiation protocols and benchmarking of single-cell computational tools. By making these data openly available, we provide a dataset relevant to research into early cell fate decisions and *in vitro* models of development.

## METHODS

### Sampling strategy

Murine embryoid body (EB) samples were generated from the 129 SvJ R1 wild-type mouse embryonic stem cell line (R1 mESCs; ATCC SCRC-1011, RRID:CVCL_2169). Cells were cultured and differentiated as described below. For hypoxic and normoxic comparisons, EBs were maintained in either 1% $O_2$ or 21% $O_2$ for the last 16 or 48 hours of differentiation, resulting in four experimental conditions in total (Figure 1).

### Steps

- Cell culture and differentiation protocol. R1 mESCs were cultured on a feeder layer of mitomycin C-inactivated mouse embryonic fibroblasts (MEFs; Sigma, M4287) in mESC medium composed of Dulbecco's modified Eagle's medium (DMEM; Gibco, 41966052) supplemented with 50 U/mL penicillin and 50 μg/mL streptomycin (Gibco, 15140122), 2 mM GlutaMAX (Sigma, G8541), non-essential amino acids (Thermo Fisher Scientific, 11140050), 0.1 mM 2-mercaptoethanol (Thermo Fisher Scientific, 21985023), 1000 U/mL ESGRO leukemia inhibitory factor (LIF; Chemicon, ESG1106), and 15% (v/v) fetal bovine serum (FBS Biowest, S1300; pre-tested for mESC culture). Medium was refreshed daily, and cells were passaged at a 1:10 ratio every other day. For EB generation, MEFs were separated from mESCs by differential adhesion to 1% gelatin in PBS for 30 min at 37 °C. mESCs were then aggregated in hanging drops (1200 cells per 20 μL drop) and cultured in mESC medium lacking LIF for 4 days. On day 4, R1-derived EBs were plated on 1% gelatin-coated dishes (28 EBs per p60 dish) to induce two-dimensional differentiation. Differentiation proceeded for a total of 8 or 10 days, with subsets of EBs transferred to hypoxia (1% $O_2$) or maintained under normoxia (21% $O_2$) for the final 16 or 48 h.



- Single-cell dissociation and library preparation. EBs were dissociated to single cells by incubation with Accumax (1 mL per p60 dish; Innovative Cell Technologies) for 30 min at 37 °C. Cells were collected by centrifugation (400 g, 5 min, 4 °C), filtered through a 70 μm strainer (Falcon, 352350), and further dissociated by gentle pipetting with wide-bore tips. Single-cell suspensions were processed for single-cell RNA sequencing (scRNA-seq) using the Chromium Single Cell 3′ Library & Gel Bead Kit (10× Genomics) following the manufacturer's protocol (10× Genomics, Pleasanton, CA, USA). Libraries were sequenced on an Illumina platform (Illumina NovaSeq 6000) according to 10× Genomics recommendations.

- Data processing and analysis. Raw sequencing reads were processed using Cell Ranger v7.1.0 (10× Genomics). Reads were demultiplexed and aligned to the mouse reference genome mm10 (refdata-gex-mm10-2020-A, GRCm38, GENCODE vM23), and gene–cell count matrices were generated. For each of the four samples, both `raw_feature_bc_matrix` and `filtered_feature_bc_matrix` HDF5 (.h5) files were produced, containing all barcodes and high-confidence cells, respectively, resulting in a total of eight HDF5 files. Default parameters were used for alignment, UMI counting, and cell calling. Downstream analyses were performed in R using the Seurat package [2–6]. Cell clusters were identified using Seurat's standard workflow, potential doublets were detected with scDblFinder [7], and hypoxia scores were calculated for each cell using ssGSEA via GSVA [8], based on the expression of eight genes (Ankrd37, Bhlhe40, Egln1, Maff, Ndrg1, Pfkfb3, and Tcaf2) comprising a hypoxia gene signature [9].

## DATA VALIDATION AND QUALITY CONTROL

Previous studies have shown that hypoxia simultaneously promotes angiogenesis and induces cell cycle arrest in endothelial cells, reflecting an adaptive balance between vascular expansion and reduced proliferation of mature endothelial populations [10]. To confirm that our system recapitulated these established biological responses, we performed several validation experiments prior to sequencing (Figure 2). Flow cytometry analysis of EBs at day 9 revealed an increased proportion of CD31[+] and CD144[+] endothelial populations under hypoxic conditions compared to normoxia, indicating enhanced endothelial differentiation. Consistently, immunofluorescence staining of CD31 at day 10 demonstrated more extensive vascular-like networks in hypoxia, with quantitative measurements showing significant increases in branch number, branch length, and total network complexity. In parallel, EdU incorporation assays revealed that hypoxia markedly reduced the proportion of cells in S-phase (Figure 3), confirming cell cycle arrest. Together, these results demonstrate that the EBs responded to hypoxia in a manner consistent with published findings and validate the biological robustness of the dataset.

To assess the technical quality of the single-cell RNA-seq dataset, we examined sequencing and mapping metrics across the four experimental conditions. A summary of key quality control statistics is provided in Table 1. Across all samples, high proportions of valid barcodes (>97%) and base quality scores (Q30 > 90% across barcode, RNA, and UMI reads) were observed, indicating robust library preparation and sequencing performance. The fraction of reads confidently mapped to the mouse genome (mm10) was consistently >92%, with 70–74% of reads aligning to exonic regions, consistent with high-quality transcript capture. Sequencing saturation ranged from 26.8% to 32.7%, reflecting adequate depth for transcriptome coverage across conditions. The number of estimated cells

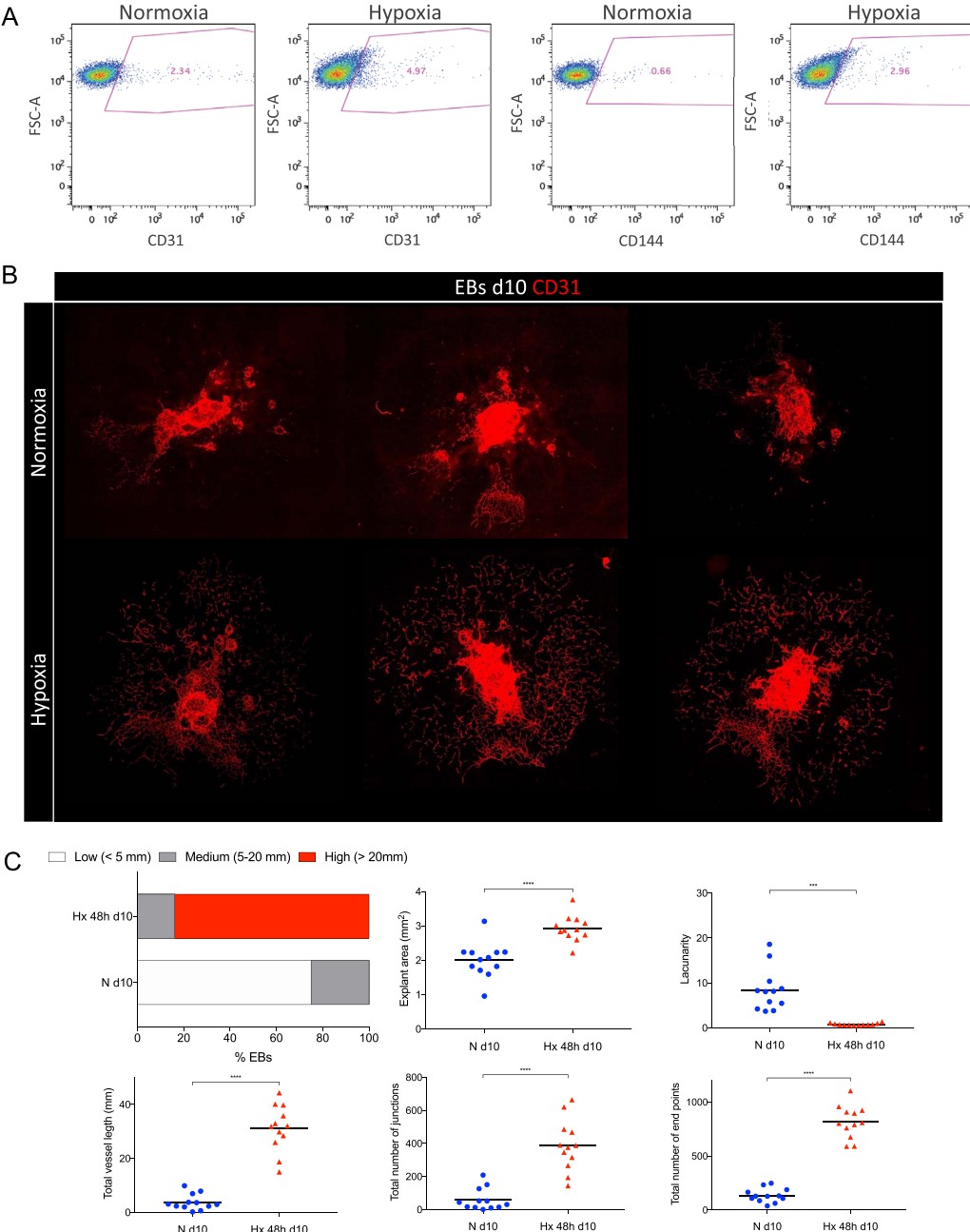

**Figure 2.** Validation of hypoxia-induced angiogenic responses in embryoid bodies. (A) Flow cytometry plots of day 9 EBs showing CD31$^+$ and CD144$^+$ endothelial populations under normoxia or hypoxia. (B) Representative immunofluorescence images of CD31$^+$ vascular networks in day 10 EBs cultured under normoxia or hypoxia. (C) Quantification of angiogenic parameters with AngioTool analysis software. The upper-left plot shows the percentage of EBs with low (<2.5 mm length, white), medium (2.5–20 mm length, grey) or high (>20 mm length, red) angiogenesis is represented for each experimental condition. A Chi-square analysis showed that the distribution of vessel length was significantly different among conditions ($X^2_2 = 161.2$, $P < 0.0001$). The remaining plots show individual AngioTool parameters for each EB: Explant area (mm$^2$), Lacunarity, Total vessel length (mm), Total number of junctions and total number of end points. Statistical significance was assessed using an unpaired two-tailed t-test with Welch's correction (** $P < 0.01$, *** $P < 0.001$, **** $P < 0.0001$).

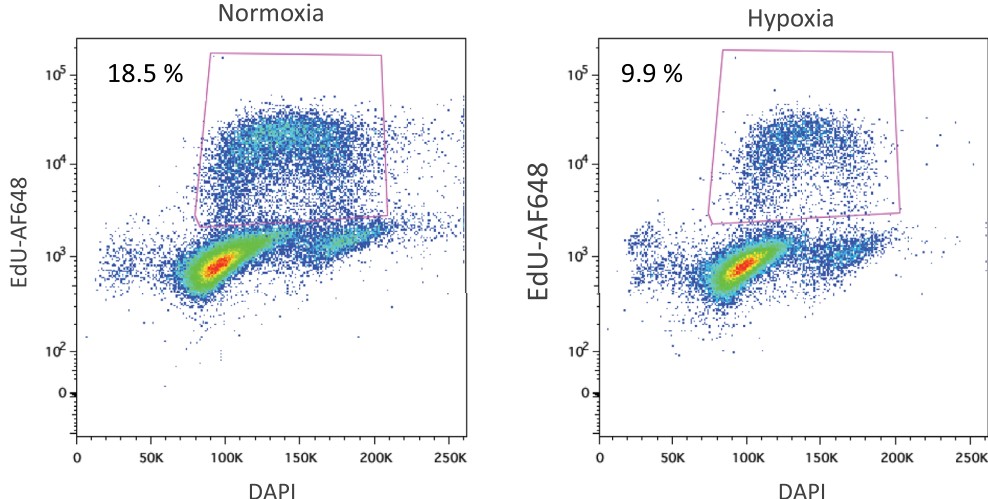

**Figure 3.** Hypoxia decreases the percentage of endothelial cells in the S-phase of the cell cycle. Representative plot of EdU-Alexa 647 vs DNA content by propidium iodide staining (PI) of day 10 EBs (normoxia or last 24 h in hypoxia). Percentage of cells in G0/G1, S, and G2/M phases of the cell cycle is shown inside the corresponding gating regions (magenta lines).

**Table 1.** Summary of sequencing statistics across conditions (Normoxia vs Hypoxia at 8 and 10 days).

| Metric | Normoxia 8d | Hypoxia 8d | Normoxia 10d | Hypoxia 10d |
|---|---|---|---|---|
| Estimated number of cells | 3374 | 3489 | 2802 | 4746 |
| Mean reads per cell | 50,502 | 43,010 | 48,338 | 31,144 |
| Median genes per cell | 5294 | 4587 | 4800 | 3104 |
| Number of reads | 170,394,384 | 150,063,590 | 135,442,885 | 147,809,980 |
| Fraction reads in cells (%) | 96.0 | 94.7 | 92.3 | 92.1 |
| Total genes detected | 25,219 | 24,024 | 23,968 | 23,695 |
| Median UMI counts per cell | 22,208 | 18,349 | 19,825 | 12,512 |
| Valid barcodes (%) | 97.5 | 97.7 | 97.4 | 97.9 |
| Sequencing saturation (%) | 30.3 | 28.5 | 32.7 | 26.8 |
| Q30 bases in barcode (%) | 94.7 | 94.5 | 94.7 | 94.7 |
| Q30 bases in RNA read (%) | 90.5 | 90.4 | 91.4 | 91.4 |
| Q30 bases in UMI (%) | 94.6 | 94.6 | 94.8 | 94.7 |
| Reads mapped to genome (%) | 94.9 | 95.9 | 95.7 | 96.1 |
| Reads mapped confidently to genome (%) | 90.9 | 92.4 | 92.2 | 92.3 |
| Reads mapped confidently to intergenic regions (%) | 3.1 | 2.4 | 3.2 | 2.7 |
| Reads mapped confidently to intronic regions (%) | 16.4 | 16.7 | 18.1 | 16.2 |
| Reads mapped confidently to exonic regions (%) | 71.3 | 73.3 | 70.9 | 73.4 |
| Reads mapped confidently to transcriptome (%) | 81.3 | 83.9 | 81.5 | 83.7 |
| Reads mapped antisense to gene (%) | 6.1 | 5.7 | 7.1 | 5.6 |

recovered per sample ranged from 2802 to 4746, with median genes detected per cell between 3104 and 5294, and median unique molecular identifiers (UMIs) per cell ranging from 12,512 to 22,208. These metrics are in line with expected values for 10× Genomics scRNA-seq datasets and confirm successful cell capture and gene expression profiling.

In order to assess scRNA-seq data quality, we first examined key library metrics across samples (Figure 4). Consistent with standard quality-control expectations, cells with higher UMI counts generally displayed lower mitochondrial read fractions, whereas low-depth cells were enriched for mitochondrial reads. In addition, the strong positive relationship

**Table 2.** Parameters set to filter low quality cells in each sample.

| Metric | Normoxia 8d | Hypoxia 8d | Normoxia 10d | Hypoxia 10d |
|---|---|---|---|---|
| Minimum counts | - | - | - | 5000 |
| Minimum features | 1200 | 1100 | 1150 | 1000 |
| mtRNA % | 10 | 10 | 10 | 10 |

**Table 3.** Number of cells on each sample before and after filtering those of low quality.

| Sample | Normoxia 8d | Hypoxia 8d | Normoxia 10d | Hypoxia 10d |
|---|---|---|---|---|
| Initial | 3390 | 3514 | 2816 | 4547 |
| Filtered | 3093 | 3220 | 2509 | 3803 |

**Table 4.** Number of cells assigned to each cell cycle phase, by sample. The percentage of cells in the sample assigned to each phase, rounded to the nearest integer, is shown in brackets.

| Phase | Normoxia 8d | Hypoxia 8d | Normoxia 10d | Hypoxia 10d |
|---|---|---|---|---|
| G0–G1 | 1078 (35%) | 2017 (63%) | 1445 (58%) | 3195 (84%) |
| S | 1078 (35%) | 377 (12%) | 589 (23%) | 149 (4%) |
| G2M | 936 (30%) | 825 (25%) | 474 (19%) | 458 (12%) |

between UMI counts and number of detected genes across all samples confirmed that increased sequencing depth was associated with greater gene detection, while cells deviating from this trend were flagged as potential low-quality outliers for removal prior to downstream analyses. To remove such low-quality cells, we applied sample-specific quality-control thresholds (Table 2) to account for variability in sequencing depth and technical noise between samples, and cells not meeting these criteria were excluded from further analysis. After filtering, we obtained the final cell numbers indicated in Table 3. Cell cycle phase assignment inferred from single-cell transcriptomic profiles revealed a marked shift toward cell cycle arrest under hypoxic conditions (Table 4). At both day 8 and day 10, hypoxia was associated with a substantial increase in the proportion of cells in G0/G1 and a concomitant reduction in S-phase and G2/M populations compared to normoxic controls. This effect was particularly pronounced at day 10, where 84% of hypoxic cells were classified as G0/G1, compared to 58% under normoxia. These results closely mirror the EdU-based measurements obtained prior to sequencing and indicate that the hypoxia-induced suppression of proliferation is robustly captured at the transcriptional level. Together, these QC results and the concordance between flow cytometry, proliferation assays, and scRNA-seq-derived cell cycle states supports the biological fidelity of the dataset and provides confidence for subsequent analyses of hypoxia-driven transcriptional programs, including dimensionality reduction, clustering, and differential expression. Moreover, the dataset has been analyzed previously in a preprint describing hypoxia-induced effects during embryoid body differentiation [11], further supporting the validity and utility of this resource.

To generate the most consistent cell clusters possible, we leveraged the stochasticity of the clustering algorithm implemented in Seurat to fine-tune the clustering resolution. Because clustering results on the same data and with the same parameters can exhibit slight run-to-run variation, we searched for parameter values that minimized this variability and thus better reflected the underlying biological diversity of the dataset. We ultimately selected a very low clustering resolution parameter, favoring the formation of relatively large, broad clusters. Given the dataset size and the expected predominance of

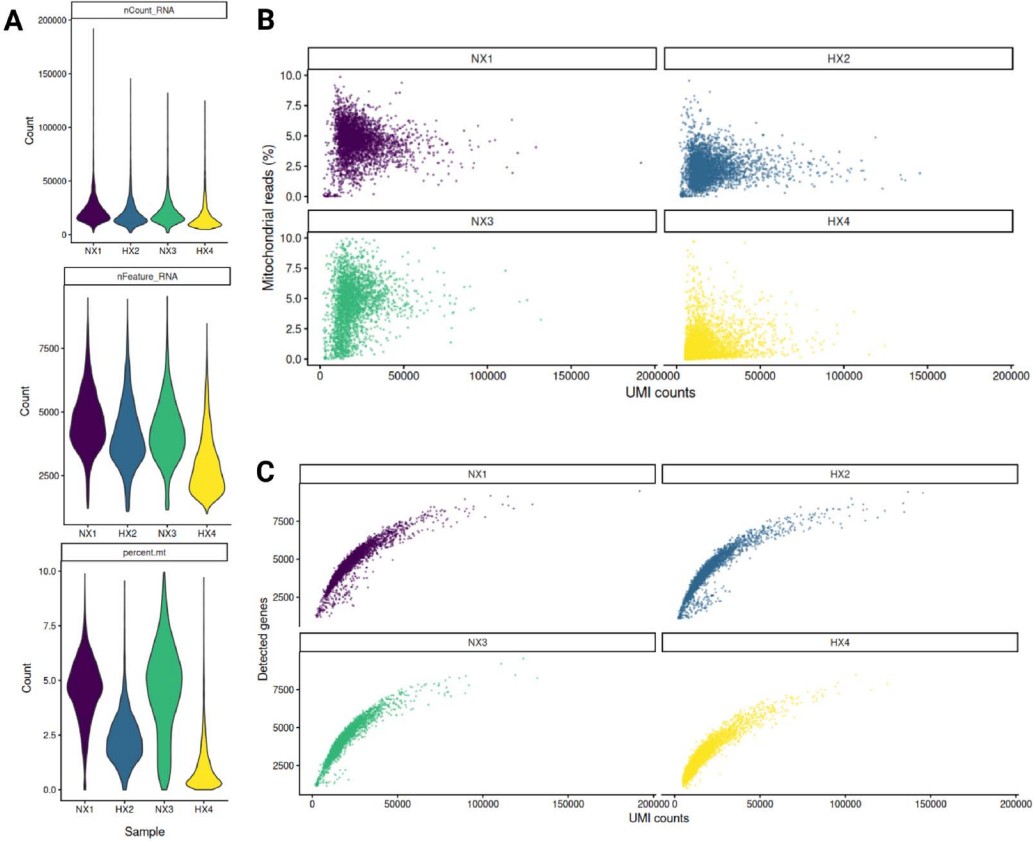

**Figure 4.** Quality control metrics of single-cell RNA-seq libraries across samples. (A) Violin plots depict the distribution of total unique molecular identifier (UMI) counts per cell (nCount_RNA), the number of detected genes per cell (nFeature_RNA), and the percentage of mitochondrial transcripts (percent.mt) for samples NX1 (Normoxia 8d), HX2 (Hypoxia 8d), NX3 (Normoxia 10d), and HX4 (Hypoxia 10d). (B) For each sample, scatter plots show the relationship between total UMI counts and the proportion of mitochondrial reads per cell. (C) Scatter plots display the number of detected genes as a function of total UMI counts for each sample.

highly related multipotent progenitors, such low resolutions help prevent the arbitrary formation of spurious, narrowly defined clusters that are difficult to reproduce and characterize with confidence. The resulting clustering, shown as a UMAP projection in Figure 5A, reveals that a small fraction of cells (10.2%) has differentiated sufficiently to be robustly separated from the main contingent of cells, whereas the remaining 89.8% of the dataset forms a more diffuse structure in which some cluster boundaries remain partially ambiguous even at this low clustering resolution. This pattern is consistent with a largely homogeneous progenitor compartment containing gradual transcriptional transitions rather than sharply delineated cell states. Such ambiguity is a known feature of sparse single-cell RNA-seq data, particularly for lowly expressed genes and rare populations [12], and has been widely discussed in the context of clustering stability and reproducibility [13, 14]. We also assessed potential doublets in the dataset, which represented only 8% of the cells and were distributed across all clusters (Figure 5B and Table 5). After removing the doublets and reclustering, we see that the clustering structure remains largely consistent with the original analysis, and the outlying subpopulations remain detectable after doublet filtering (Figure 5C). In addition, we evaluated the hypoxic transcriptional signature using

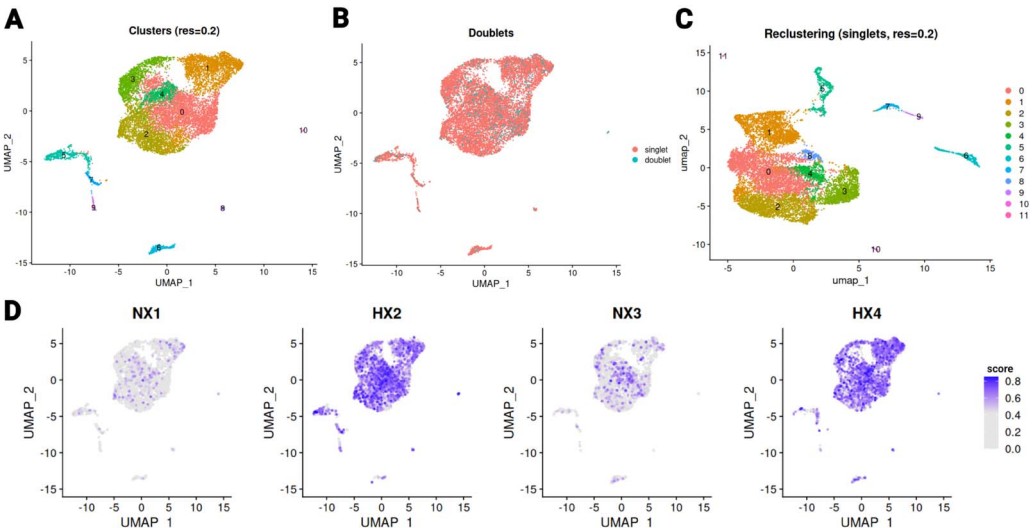

**Figure 5.** Exploration of the single-cell RNA-seq data. (A) Two-dimensional UMAP embedding of all filtered cells, colored by cluster assignment using Seurat clustering (resolution = 0.2). (B) Two-dimensional UMAP embedding of all filtered cells, colored by singlets versus doublets. (C) Two-dimensional UMAP embedding of singlet cells after removal of predicted doublets. Cells are colored according to clusters obtained after reclustering following doublet filtering. (D) Two-dimensional UMAP embedding of all filtered cells, colored by the hypoxia score calculated using GSVA based on the expression of Ankrd37, Bhlhe40, Egln1, Maff, Ndrg1, Pfkfb3, and Tcaf2 as a hypoxia gene signature.

**Table 5.** Distribution of singlets and doublets across clusters.

| Cluster | Singlets | Doublets |
|---|---|---|
| 0 | 3491 | 390 |
| 1 | 2461 | 269 |
| 2 | 2404 | 97 |
| 3 | 1314 | 73 |
| 4 | 757 | 74 |
| 5 | 518 | 39 |
| 6 | 338 | 20 |
| 7 | 178 | 11 |
| 8 | 68 | 9 |
| 9 | 69 | 4 |
| 10 | 17 | 20 |

the expression of eight genes previously shown to be associated with hypoxia (Ankrd37, Bhlhe40, Egln1, Maff, Ndrg1, Pfkfb3, and Tcaf2) [9], confirming that cells from the hypoxia samples exhibited higher hypoxia scores compared to normoxia (Figure 5D). Together, these observations support a conservative clustering strategy that captures robust, higher-level cellular structure while avoiding over-partitioning driven by technical noise.

## REUSE POTENTIAL

This dataset can serve as a valuable reference for multiple applications. It enables comparative studies of early differentiation under varying oxygen conditions and benchmarking of single-cell analysis pipelines. The resource is suitable for integration with other single-cell transcriptomic studies of differentiation and development, enabling comparative and cross-condition analyses using established integration frameworks [15]. In

this regard, the dataset complements existing single-cell and spatiotemporal transcriptomic resources for developmental biology, such as those generated for murine cardiac development [16]. Furthermore, it may be used in training or testing computational models for cell type classification and transcriptional heterogeneity. Beyond developmental biology, the data may also support broader research on hypoxia-related mechanisms relevant to disease, regeneration, and cellular stress responses.

## DATA AVAILABILITY

The dataset supporting the results of this article is available in the Gene Expression Omnibus (GEO) repository under accession number GSE226454. The processed data for generating figures is deposited in Zenodo [17].

## LIST OF ABBREVIATIONS

EB, Embryoid body; GEO, Gene Expression Omnibus; scRNA-seq, Single-cell RNA sequencing; UMI, Unique molecular identifier.

## DECLARATIONS

### Ethical Approval

Not applicable.

### Consent for publication

Not applicable.

### Competing interests

The authors declare that they have no competing interests.

### Authors' contributions

Study conceptualization: LP, BJ; experimental design: BJ, LP, BAI; data generation: BAI, BJ; bionformatic analysis: LPS, YB; writing original draft: YB; funding acquisition: LP, BJ.

### Funding

This research was funded by Ministerio Ciencia e Innovación (MCIN/AEI/10.13039/501100011033 "FEDER: A way of making Europe" and "NextGenerationEU"/PRTR, Spain) grant number PID2020-118821RB-I00 awarded to LP and BJ, by PRE2021-098587 funded by MCIN/AEI/10.13039/501100011033 and by FSE+ awarded to LP, BJ and YB, and Consejería de Ciencia, Universidades e Innovación de la CAM (Madrid, Spain) grant number IND2019/BMD-17134, awarded to LP and LPS, and grant number P2022/BMD-7224 (INSPIRA-CM) awarded to LP.

## ACKNOWLEDGEMENTS

Not applicable.

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
