## [Editor Report]

Editor’s AssessmentThe manuscript is ready for fomal acceptance.Editor’s AssessmentThe manuscript is ready for fomal acceptance.

---

## [Reviewer Report]

Indicate in the comments box below whether you are happy with the changes made or if the manuscript is unacceptable.Comments on revised manuscriptThe revised version has largely addressed the previous concerns. The methods are described in greater detail, the data presentation is more comprehensive, and the overall quality of the manuscript has improved substantially. However, I have some concerns regarding the cell clustering shown in Figure 5. Several clusters (e.g., clusters 6, 8, and 10) appear to be strong outliers. It would be helpful to further examine these subpopulations, for example by assessing whether they are influenced by batch effects and whether batch-effect correction using appropriate software is necessary, although it is also possible that these subclusters are biologically meaningful. In addition, performing doublet detection and filtering prior to re-clustering the cells would likely be more appropriate, as some outlier subclusters (e.g., cluster 10) might disappear after these steps.

---

## [Reviewer Report]

Reviewer name and names of any other individual's who aided in reviewer Gerardo CorderoDo you understand and agree to our policy of having open and named reviews, and having your review included with the published papers. (If no, please inform the editor that you cannot review this manuscript.)YesIs the language of sufficient quality?YesPlease add additional comments on language quality to clarify if needed
Are all data available and do they match the descriptions in the paper? YesAdditional CommentsAre the data and metadata consistent with relevant minimum information or reporting standards? See GigaDB checklists for examples <a href="http://gigadb.org/site/guide" target="_blank">http://gigadb.org/site/guide</a>YesAdditional CommentsIs the data acquisition clear, complete and methodologically sound?YesAdditional CommentsIs there sufficient detail in the methods and data-processing steps to allow reproduction?YesAdditional CommentsIs there sufficient data validation and statistical analyses of data quality? YesAdditional CommentsIs the validation suitable for this type of data?YesAdditional CommentsThe experimental effect was validatedIs there sufficient information for others to reuse this dataset or integrate it with other data?YesAdditional CommentsAny Additional Overall Comments to the AuthorMinor comments 1)In the second line of the abstract please change the proposition ‘while' to ‘although’ 2)Which Illumina platform did you use? 3)Do you have quality control metrics for mitochondrial contamination? This can be used as an indicator of a reduction of cell viability during processing.RecommendationMinor Revision

---

## [Reviewer Report]

Reviewer name and names of any other individual's who aided in reviewer Wei ZhangDo you understand and agree to our policy of having open and named reviews, and having your review included with the published papers. (If no, please inform the editor that you cannot review this manuscript.)YesIs the language of sufficient quality?YesPlease add additional comments on language quality to clarify if needed
Are all data available and do they match the descriptions in the paper? YesAdditional CommentsAre the data and metadata consistent with relevant minimum information or reporting standards? See GigaDB checklists for examples <a href="http://gigadb.org/site/guide" target="_blank">http://gigadb.org/site/guide</a>YesAdditional CommentsIs the data acquisition clear, complete and methodologically sound?YesAdditional CommentsIs there sufficient detail in the methods and data-processing steps to allow reproduction?NoAdditional CommentsThe experimental procedures are generally clear; however, the data analysis requires further improvement. More detailed descriptions of the data processing and analysis steps are needed. For example, specific parameters used in Cell Ranger should be explicitly reported. Additional downstream processing using commonly adopted tools such as Seurat to generate cell clustering results would be beneficial. This would not only provide an extra layer of data quality assessment, but also facilitate data reuse by enabling users to work directly with processed datasets without the need to perform a full reanalysis.Is there sufficient data validation and statistical analyses of data quality? NoAdditional CommentsThe authors provide both biological and technical validations supporting the robustness of the dataset, and standard single-cell RNA-seq quality metrics indicate high technical quality. However, as noted above, additional downstream analyses could further characterize data quality, for example by estimating the proportion of doublets and assessing the fraction of cells with high mitochondrial gene content, among other commonly used metrics. The authors note that the data have been analyzed in a preprint manuscript; including more detailed analyses in the present manuscript would further strengthen the value of this data release.Is the validation suitable for this type of data?NoAdditional CommentsWhile the authors present solid biological and technical validations, and independent assays demonstrate hypoxia responses consistent with previous studies, the sequencing data represent the core contribution of this manuscript. Additional analyses leveraging the single-cell transcriptomic data to directly examine angiogenesis or endothelial expansion would further strengthen the validation and enhance the value of the dataset for reuse.Is there sufficient information for others to reuse this dataset or integrate it with other data?NoAdditional CommentsAlthough the reuse potential is clearly articulated, providing more concrete details on the structure and contents of the deposited data—such as the number of samples and file types—would further facilitate data reuse and integration.Any Additional Overall Comments to the AuthorThis manuscript describes a well-designed single-cell RNA-seq dataset generated from murine embryoid bodies differentiated under controlled hypoxic and normoxic conditions. The experimental procedures are generally clear and methodologically sound, and the dataset has potential value as a community resource. However, several issues should be addressed: 1） More detailed downstream analyses would strengthen the manuscript and better demonstrate the utility and quality of the dataset. 2）The overall data size is relatively limited, comprising only four experimental conditions/time points, which may restrict its broader applicability. 3）The authors state that these sequencing data have already been used in a preprint manuscript by the same group. It is therefore unclear whether the dataset remains appropriate for publication in this journal as a standalone Data Release.RecommendationMajor Revision